

# A storage-efficient ensemble classification using filter sharing on binarized convolutional neural networks

HyunJin Kim[1], Mohammed Alnemari[2] and Nader Bagherzadeh[3]

[1] School of Electronics and Electrical Engineering, Dankook University, Yongin-si, Gyeonggi-do, South Korea
[2] School of Electrical and Computer Science, University of California, Irvine, Irvine, California, United States
[3] Department of Electrical Engineering and Computer Science, University of California, Irvine, Irvine, California, United States

## ABSTRACT

This paper proposes a storage-efficient ensemble classification to overcome the low inference accuracy of binary neural networks (BNNs). When external power is enough in a dynamic powered system, classification results can be enhanced by aggregating outputs of multiple BNN classifiers. However, memory requirements for storing multiple classifiers are a significant burden in the lightweight system. The proposed scheme shares the filters from a trained convolutional neural network (CNN) model to reduce storage requirements in the binarized CNNs instead of adopting the fully independent classifier. While several filters are shared, the proposed method only trains unfrozen learnable parameters in the retraining step. We compare and analyze the performances of the proposed ensemble-based systems depending on various ensemble types and BNN structures on CIFAR datasets. Our experiments conclude that the proposed method using the filter sharing can be scalable with the number of classifiers and effective in enhancing classification accuracy. With binarized ResNet-20 and ReActNet-10 on the CIFAR-100 dataset, the proposed scheme can achieve 56.74% and 70.29% Top-1 accuracies with 10 BNN classifiers, which enhances performance by 7.6% and 3.6% compared with that using a single BNN classifier.

## INTRODUCTION

Deep Neural Networks (DNNs) have been attractive in many fields, including computer vision, data mining, natural language processing, speech recognition, etc. Notably, CNNs show many outstanding performances in applications in the field of computer vision (*Alzubaidi et al., 2021*). Filter parameters in convolutional layers are automatically trained based on a target dataset. Simple CNNs such as LeNet-5 (*LeCun et al., 1998*) can be implemented on embedded devices. However, large memory requirements and computational power consumption of state-of-the-art CNN models preclude their use in lightweight systems.

Corresponding author
HyunJin Kim,
hyunjin2.kim@gmail.com

The quantization scheme utilizes the error-resilience of CNNs by sacrificing the precision of model parameters, which reduces memory requirements and power consumption (*Wu et al., 2016*). Notably, a BNN quantizes activations and weights in convolutional layers into +1 or −1 (*Courbariaux et al., 2016*). The floating-point multiplication and accumulation are replaced by the binary bitwise XNOR and bit count operations, respectively, which can save memory requirements by 32× and computational power 58× (*Rastegari et al., 2016*). Also, the simplified operations of BNNs make it possible to implement the XNOR-Net model on the CPU-based embedded system or Field-Programmable Gate Array (FPGA) (*Zhou, Redkar & Huang, 2017*; *Yi, Xiao & Yongjie, 2018*; *Liang et al., 2018*). Although there are many benefits in reducing hardware costs, performance is degraded compared with the CNN based on floating-point format and operations.

Generally, the power sourcing status of an embedded system can be changeable. When a battery pack provides its power, the system's energy-saving is essential to increase the time when the system is on. On the other hand, if an external power source or energy harvesting environment is available, the system can switch into high-speed mode by consuming enough power. If the system can have different trained models depending on its power source status, it can provide high performance when power is enough. However, lightweight systems could not contain different types of models due to their storage limitations.

An ensemble-based system can improve the performance of CNNs by averaging the classification results from different models (*Hansen & Salamon, 1990*). Each model acts as a single base classifier, and the combined prediction of multiple base classifiers is provided from the ensemble-based system with CNNs. In the same manner, ensemble BNNs can obtain better classification results using multiple models (*Vogel et al., 2016*; *Zhu, Dong & Su, 2019*), which increase the regularization of target solutions and enhance the inference accuracy. The ensemble in *Vogel et al. (2016)* stores weights of base classifiers derived from a BNN model by applying the stochastic rounding to each real-valued weight multiple times. The study in *Zhu, Dong & Su (2019)* shows the trade-offs on the number of classifiers with BNNs. The methods in *Vogel et al. (2016)* and *Zhu, Dong & Su (2019)* increase the inference accuracy using multiple base classifiers, so that memory requirements are proportional to the number of weight files in base classifiers. They could not be suitable for the embedded system with limited storage resources.

Our study focuses on a method for overcoming the storage cost limitation required by BNN ensembles. The storage costs required by BNN ensemble models can be reduced by sharing the filters in convolutional layers. While base classifiers share the filter weights from the convolutional layers of a pretrained BNN model, the affine parameters of the batch normalization layer and weights of the fully connected layer are only retrained for ensemble-based systems.

We summarize our contributions as follows:

- We propose an ensemble method based on shared filter weights that reduce the amount of storage required for ensemble-based systems.

- In the proposed ensemble system, we show the scalability with the number of base classifiers in terms of classification accuracy.
- We adopt various ensemble methods and compare the details of each method in our evaluations. Details of experimental environments are described to apply the proposed method based on several BNNs. Notably, with a binarized ResNet-20 model (*Rastegari et al., 2016*) on the CIFAR-100 dataset (*Krizhevsky, Nair & Hinton, 2014*), the proposed scheme can achieve 56.74% Top-1 accuracy with 10 BNN classifiers, which enhances the performance by 7.58% compared with that using a single BNN model. For the state-of-the-art ReActNet-10 (*Liu et al., 2020*) on the CIFAR-100 dataset, our method produces Top-1 accuracy improvements up to 3.6%.

In our experiments, we apply different ensemble schemes to binarized ResNets from XNOR-Net (*Rastegari et al., 2016*), Bi-Real-Net (*Liu et al., 2018*), and ReActNet (*Liu et al., 2020*). When base classifiers share filter weights, our experimental data shows performance enhancements. The fusion scheme is the best of ensembles when using binarized ResNet-20. The bagging scheme shows good performance enhancements when using binarized ResNet-18. Our method are evaluated with Bi-Real-Net (*Liu et al., 2018*) and ReActNet (*Liu et al., 2020*). Notably, although the ensemble-based systems share filter weights, they can show the scalability with the number of base classifiers.

This paper is structured as follows: in preliminary section, we introduce several related works and explain BNNs in detail, as well as various ensemble methods. Then, this paper describes our motivation and the proposed ensemble-based systems. Finally, experimental results and analysis show that the proposed method improves the inference accuracy with scalability.

## PRELIMINARIES

### Low-cost neural networks

Particularly for low-cost edge devices, the ultimate design goal is to create low-cost neural network models. Low-cost neural network models have a small number of multiply-accumulate operations, which makes them efficient in terms of storage and computational costs, as well as easy to deploy on the edge. Besides, lightweight data formats and their operations for low-cost neural networks have been developed along with training methods based on them.

Pruning is a well-studied method for reducing both computational and storage costs of DNNs by reducing the number of multiply-accumulate operations. In the early stages of applying pruning on DNN models, connections can be pruned based on the lowest saliency (*LeCun, Denker & Solla, 1990*). The saliency term comes from computing the Hessian matrix or inverse Hessian matrix for every parameter, as shown in the Optimal Brain Damage (OBD) and Optimal Brain Surgeon (OBS) methods, respectively (*LeCun, Denker & Solla, 1990*; *Hassibi & Stork, 1993*). However, for DNN models such as AlexNet (*Krizhevsky, Sutskever & Hinton, 2012*) and VGG (*Chatfield et al., 2014*), it is not plausible to compute the Hessian matrix or inverse Hessian matrix for every parameter. In the deep compression method of *Han, Mao & Dally (2015)*, a certain threshold can be used to

remove connections below a certain threshold. Although floating-point operations for convolutional layers could be dramatically compressed, the implementation of pruning DNNs require special libraries such as Sparse Basic Linear Algebra Subprograms (BLAS) (*Han, Mao & Dally, 2015*) or special hardware accelerators to deal with sparse matrices (*Han et al., 2016*). Methods in *Li et al. (2016)*, *Luo, Wu & Lin (2017)*, and *He et al. (2019)* prune filters of CNNs so that those do not require specialized libraries or hardware blocks. However, due to their coarse granularity of compression, the ratio of reduced floating-point operations cannot be significant.

Quantization methods reduce overall costs by adopting lightweight data formats and their operations. Mainly, quantization focuses on how many bits are needed to represent DNN model parameters. When quantizing DNN models, weights, activations, or inputs from real-valued format (*e.g.*, 32-bit floating point) are converted into one or several lower-precision formats. Notably, 16-bit fixed-point (*Gupta et al., 2015*), 8-bit fixed-point (*Han, Mao & Dally, 2015*; *Wu et al., 2018*; *Wang et al., 2018*), and logarithmic (*Miyashita, Lee & Murmann, 2016*) formats are adopted in existing quantization CNN models. The hardware complexity of the multiply-accumulate operation in convolution can decrease depending on the degree of quantization.

In post-training quantization, a DNN model is trained based on a real-valued format for producing its pretrained model. Then, quantized parameters of the pretrained model are adopted for realizing low-cost inference stages. However, the training loss does not consider quantization errors during training so that highly quantized model could not provide acceptable classification results. On the other hand, quantization-aware training considers quantization errors during training steps. Although training loss from the quantization errors requires long training time, its trained model can provide better classification results compared with those using post-training quantization.

BNNs produce highly quantized models by only using 1-bit format to represent DNN parameters. Generally, quantization errors from their binarization are considered during training. In the following, BNNs are fully explained.

## Binary neural networks

In CNNs, as the numbers of layers and channels increase, there are large memory requirements for storing model parameters. Also, tremendous multiply-accumulate operations in convolutions need large computational power consumption. A low-cost embedded inference system could not have enough memory units to store 32-bit floating-point (fp32) model parameters. Besides, parallel fp32 units are not equipped considering their low-cost implementation. The quantization in CNNs reduces memory requirements and power consumption by adopting inaccurate data computation (*Courbariaux, Bengio & David, 2015*; *Hubara et al., 2016*; *Hubara et al., 2017*; *Rastegari et al., 2016*). Notably, BNNs can quantize both weights and activations of CNNs into −1 and +1 in their forward paths, significantly decreasing storage resource requirements for saving parameters. The multiply-accumulate operation in convolutions can be approximated using XNOR and bit count operations, thus reducing computational resources. For

example, in *Rastegari et al. (2016)*, a BNN can achieve ~32× memory efficiency and ~58× speedup on a single core device, compared with its corresponding fp32-based CNN.

In *Rastegari et al. (2016)*, the dot product in a convolution is approximated as follow: We define the real-valued activation and weight filter as $\mathbf{X}$ and $\mathbf{W}$. In convolutional layers of BNNs, the dot product is approximated between $\mathbf{X}, \mathbf{W} \in \mathbf{R}^n$ such that $\mathbf{X}^T\mathbf{W} \approx \gamma\mathbf{H}^T\alpha\mathbf{B} = k\mathbf{H}\odot\mathbf{B}$, where $\mathbf{H}, \mathbf{B} \in \{+1, -1\}^n$ and $\alpha, \gamma \in \mathbf{R}^+$. In other words, $\mathbf{H}$ and $\mathbf{B}$ denote the binary activation and filter. Terms $\alpha$ and $\gamma$ are scaling factors for weights and activations, respectively. Symbol $\odot$ denotes the dot product of vectors $\mathbf{H}$ and $\mathbf{B}$ using XNOR-bitcount operations. Element-wise matrix product is obtained by multiplying vector $\mathbf{H}\odot\mathbf{B}$ with $k = \gamma\alpha$. Whereas term $k$ is deterministically calculated in *Rastegari et al. (2016)*, $k \in \mathbf{K}$ is a learnable affine parameter in *Bulat & Tzimiropoulos (2019)* and *Kim (2021)*. The dot product for reducing quantization errors can be optimized as:

$$\alpha, \mathbf{B}, \gamma, \mathbf{H} = \underset{\alpha, \mathbf{B}, \gamma, \mathbf{H}}{\operatorname{argmin}} \|\mathbf{X}\odot\mathbf{W} - \gamma\alpha\mathbf{H}\odot\mathbf{B}\|. \tag{1}$$

The binarization of $\mathbf{X}$ using *sign*() function is applied to deterministically quantize each feature $x \in \mathbf{X}$ in the binary activation layer. A binary activation is formulated as:

$$x \in \mathbf{X}, sign(x) = \begin{cases} +1 & if \ x \geq 0, \\ -1 & else. \end{cases} \tag{2}$$

The derivative of *sign*() function contains the $\delta$ function, so it is approximated in the backward path during training a BNN model. BNNs in *Rastegari et al. (2016)* and *Courbariaux, Bengio & David (2015)* adopt the *straight-through-estimator* approach in the approximation of its derivative. Unlike the conventional convolutional layer, the binarized activation is the input to the convolutional layer.

The batch normalization (BN) (*Ioffe & Szegedy, 2015*) layer is placed before the binary activation layer. The BN is formulated as:

$$\hat{x}_i \rightarrow \frac{x_i - \mu_\beta}{\sqrt{\sigma_\beta^2 + \varepsilon}}, \tag{3}$$

where $\mu_\beta$ and $\sigma_\beta$ are the mini-batch mean and variance for a channel. Iterative mini-batches from the previous layer's outputs are used in the training of $\mu_\beta$ and $\sigma_\beta$. A convolution output $x_i$ means each element in a channel. A small constant $\varepsilon$ prevents the division by zero. After the normalization, the BN layer scales and shifts the normalized feature $\hat{x}_i$ into $x_i$ in a channel, which can be equated as:

$$x_i \rightarrow \lambda\hat{x}_i + \beta, \tag{4}$$

where the affine parameters $\lambda$ and $\beta$ are learnable during the CNN training.

The BN layer can change the range of the convolution output distribution, and the adjusted convolution output is used as the input to the binary activation layer. Figure 1 illustrates the layer connection for a convolutional layer, where *Conv* and *BinConv* denote the conventional fp32-based convolutional and binarized convolutional layers,

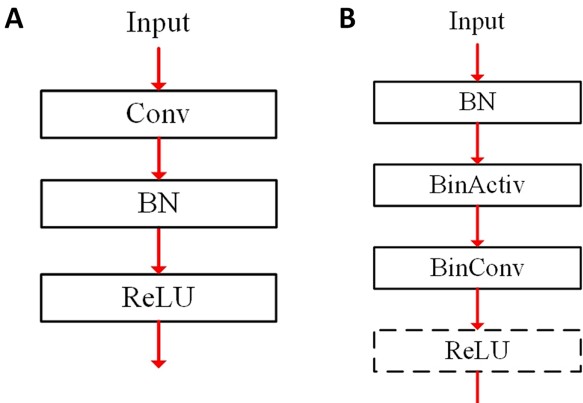

**Figure 1** **Layer connection for convolutional layers: (A) conventional fp32 (B) BNN.**

respectively. Besides, the term *BN* and *BinActiv* mean the BN and binary activation layers. In a conventional fp32-based CNN of Fig. 1A, the convolutional layer outputs go towards the next BN layer, and the BN layer outputs go towards the activation layer such as *ReLU*. Fig. 1B shows the layer connection of a binarized convolution in *Rastegari et al. (2016)*. The BN layer is located before the BinActiv layer to adjust features with its learnable parameters. The features binarized from the BinActiv layer go towards the BinConv layer. It is noted that the output format of the BinConv layer becomes fp32 after its scaling.

During BNN training, the fp32 format is generally adopted to update the weights of the binarized convolutional layer in the backward path. Besides, the parameters of the BN layer also use fp32 format. Because the derivative of $sign(x)$ function is the $\delta$ function, the derivative of the $sign(x)$ function in the binary activation layer should be approximated. Generally, a *straight-through-estimator* (*Rastegari et al., 2016*; *Courbariaux et al., 2016*) for this approximation is formulated as follows:

$$\frac{\partial sign(x)}{\partial x} = \begin{cases} 1 & if \ -1 \leq x \leq 1, \\ 0 & else. \end{cases} \tag{5}$$

## Ensemble-based systems

Ensemble methods, also known as ensemble learning, are a powerful tool to improve deep neural network models and provide better model generalization. An ensemble-based system is defined as the implementation of ensemble learning. An ensemble-based system combines multiple machine learning algorithms or multiple different models to produce better performance than those adopting a single algorithm or model. An ensemble-based system consists of multiple base estimators that are combined to form a strong estimator (*Hansen & Salamon, 1990*). Ensemble methods include fusion, voting, bagging, gradient boosting, and many others. Finding the best model that yields the least error in the search space is an important problem in statistical learning, which is the goal of machine learning. This is a hard work because datasets are always smaller than the

search space. Researchers have discovered a way to mitigate this by using various ensemble methods. Voting reduces the risk of selecting a bad model. Bagging and boosting with different starting points result in a better approximation. Fusion expands model's function space (*Ganaie et al., 2021*). Furthermore, when homogeneous estimators of the same type are built with a different activation function or initialization method, homogeneous ensemble can increase diversity while decreasing correlation (*Maguolo, Nanni & Ghidoni, 2021*).

In CNNs, estimators are mainly used for classifications so that base classifiers are combined to produce one strong classifier for predicting the class for a given sample. There are different libraries that help to assist building ensemble methods with state-of-art DNNs. In this paper, we use *Ensemble-Pytorch* library (*Xu, 2020*), which is an open source library that supports different type of the ensemble methods.

Fusion and voting are basic ensemble-based schemes. In the fusion-based ensemble, the averaged prediction from base classifiers is used to calculate the training loss. When $M$ base classifiers $\{e^1, e^2, \ldots, e^m, \ldots, e^M\}$ are adopted, the output can be $\mathbf{o_i} = \frac{1}{M} \sum_{m=1}^{M} \mathbf{o_i}^m$ for a given sample $\mathbf{s_i}$. In the fusion, when $y_i$ is the target output for $\mathbf{s_i}$, its loss function is $L(\mathbf{o_i}, y_i)$. On data batch $D$, its training loss can be $\frac{1}{D} \sum_{i=1}^{D} L(\mathbf{o_i}, y_i)$. The training loss is used to update parameters of all base classifiers.

When using a voting-based ensemble, each base classifier is created independently. With $M$ base classifiers, a base classifier $e^m$ can be trained without considering other base classifiers. In inference, hard voting gathers so-called *votes* from base classifiers and selects the class with majority vote. For example, let us assume that there are two classes ($\{dog, cat\}$) in a dataset and three base classifiers ($\{e^1, e^2, e^3\}$) in a voting-based ensemble. When classifiers $e^1$ and $e^2$ classify a sample into *dog*, *dog* is voted on the classification. Even if classifier $e^3$ votes for *cat*, *dog* is selected by a majority vote. On the other hand, soft voting sums the prediction probabilities from classifiers and then averages the summed values. The class with a high probability is selected in soft voting. For example, let us assume that a base classifier $e^m$ outputs its predicting probabilities of classifying a sample for a set of classes $\{dog, cat\}$, which is formulated as $P(e^m) = \{P_{dog}, P_{cat}\}$. When two base classifiers have $P(e^1) = \{0.7, 0.3\}$ and $P(e^2) = \{0.2, 0.8\}$, their averaged probabilities can be $P_{avg} = \left\{\frac{0.7+0.2}{2}, \frac{0.3+0.8}{2}\right\}$. Therefore, the sample is classified into *cat* in this soft voting.

In the bagging-based ensemble (*Breiman, 1996*), the subsampling with replacement produces multiple datasets to train each base classifier. In *Xu (2020)*, different data batches for each base classifier are sampled with replacement and used to train base classifiers independently.

The boosting-based ensemble trains base classifier sequentially, where a base classifier is trained considering the errors from the previously trained base classifier. In the snapshot ensemble (*Huang et al., 2017*), unlike other ensemble-based systems, only one model exists, and the model parameters are collected at each minima during its training. Therefore, multiple base classifiers are obtained from the different parameters of the during training.

Notably, the base classifier can be base classifiers for the image classification. Generally, when base classifiers are adopted, the averaging of base classifiers is performed on the predicted probabilities of target classes *via* a softmax function as:

$$softmax(\mathbf{z}_i^j) = \frac{e^{z_i^j}}{\sum_{k=1}^{K} e^{z_k^j}}, \tag{6}$$

where term $K$ is the number of classes. Term $z_i^j$ is an element of the input vector $\mathbf{z}$ on the $j$-th base classifier so $\mathbf{z}^j = (z_i^j, ..., z_K^j) \in \mathbb{R}^K$. In inference, predictions from the base classifiers are averaged.

There have been several ensemble-based systems using BNNs. The ensemble-based system in *Vogel et al. (2016)* applies the stochastic rounding to a real-valued weight to get its binary weight. The stochastic rounding of a weight $w$ can be performed in Eq. (7) as:

$$sr(w) = \begin{cases} \lfloor w \rfloor & \text{with probability } 1 - (w - \lfloor w \rfloor), \\ \lfloor w \rfloor + 1 & \text{with probability } w - \lfloor w \rfloor. \end{cases} \tag{7}$$

This system performs a kind of soft voting with base classifiers that contain different binary weight files from one high-precision neural network. Each inference evaluation with a binary weight file could be considered as a base classifier, The ensemble-based system averages prediction probabilities to enhance the classification accuracy. Although this ensemble-based system lowers the classification variance of the aggregated classifiers (*Vogel et al., 2016*), its target system should store multiple weight files. Binary Ensemble Neural Network (BENN) (*Zhu, Dong & Su, 2019*) adopts bagging and boosting strategies to obtain multiple models to be aggregated in *Zhu, Dong & Su (2019)*. These sophisticated ensemble methods improve inference accuracy, but multiple weights should be stored as well, which is a significant burden when using BNNs.

# PROPOSED METHOD

## Motivations

BNNs reduce both storage requirements and computation time using binary weights and related binary bitwise operations. Therefore, BNNs are suitable for power-hungry embedded systems. However, their inference accuracies are degraded compared with those of real-valued CNNs.

A power-hungry embedded system has different power source status. For example, if an external power source is connected, enough power is available to the system. Sometimes, even if a system consumes more power, it requires high inference accuracy. If a system has energy harvesting equipment, the system can have other power source statuses. When the system can gather enough power, there is no need to continue the low power mode. Depending on its power source status, it is necessary to provide adequate inference accuracy by adjusting the amount of computation.

We note that the ensemble-based system using BNNs can provide this trade-off. In the ensemble-based system using BNNs, multiple BNN models are aggregated to produce better classification results, where each trained model can act as a base classifier. Whereas

only one BNN model can be used in a low power mode, multiple BNN models can be aggregated when power source is enough. When using multiple BNN models, multiple sets of model parameters should be stored for providing multiple base classifiers. However, if storage resources are limited in the system, this ensemble-based system is not applicable. Except for the voting-based ensemble, the ensemble-based systems introduced in Preliminaries cannot train base classifiers independently. In other words, when varying the number of aggregated classifiers in the ensembles, each ensemble gets a different set of learnable model parameters.

Although there are significant advantages of BNNs, the increase in storage requirements for the ensemble-based system can limit their applications. We are motivated that if parameters are shared between BNN models, the storage resource limitation can be alleviated in the ensemble. We consider the characteristics of CNNs to determine which parameters are shared. In image classification using CNNs, convolutional filter weights are automatically learned during their training process. Each filter extracts the abstract meaning from features, which are connected by performing hierarchical convolutions. We think that classifiers share the abstraction tools. Therefore, filters could be shared as the tools, extracting the abstraction from input features in the proposed method. On the other hand, the obtained abstractions can be differently adjusted and connected in each classifier. In the following, details of the proposed method are explained.

## Proposed ensemble-based system using BNNs

Figure 2 illustrates the concept of the proposed ensemble-based system. Multiple base classifiers are constructed using a given pretrained model. In Fig. 2, two base classifiers $e_1$ and $e_2$ have an identical structure with the pretrained model. Between base classifiers, the parameters of the BinConv layers in the same position can be the same to share filter weights. When sharing the filter weights, the filter weights are based on those of the pretrained model. When initializing base classifiers, the filter weights of BinConv layers are adopted. During the retraining process, they are *frozen* in all base classifiers without updating. Therefore, the filter weights of the BinConv layers in each base classifier maintain the initial values and do not change during the retraining process.

We define the binary weight filter $\mathbf{B}^m$ from $m$-th base classifier, where $m \in \{i \in \mathbb{N} \mid i \leq M\}$. When the filter weights are reused, $\mathbf{B}^m = \mathbf{B}$. Term $\mathbf{H}^m$ denotes the binary activation from $m$-th base classifier. Thus, the dot product for a base classifier is optimized as:

$$\alpha, \mathbf{B}, \gamma^m, \mathbf{H}^m = \underset{\alpha, \mathbf{B}, \gamma^m, \mathbf{H}^m}{\operatorname{argmin}} \|\mathbf{X}^m \odot \mathbf{W} - \gamma^m \alpha \mathbf{H}^m \odot \mathbf{B}\|. \tag{8}$$

On the other hand, the parameters of BN layers in the same position are different in base classifiers. Formally, $x_i^m \rightarrow \lambda^m \hat{x}_i^m + \beta^m$, where $x_i^m \in \mathbf{X}^m$. Parameters $\lambda^m$ and $\beta^m$ for $e^m$ are learnable in the retraining process. The scaling and shifting with parameters $\lambda^m$ and $\beta^m$ for each channel can adjust the normalized features to optimize the ensemble-based system in a base classifier. In a pretrained model, when an activation value $x_i$ becomes close to zero, its quantization error is maximized, which could produce large bias and variance

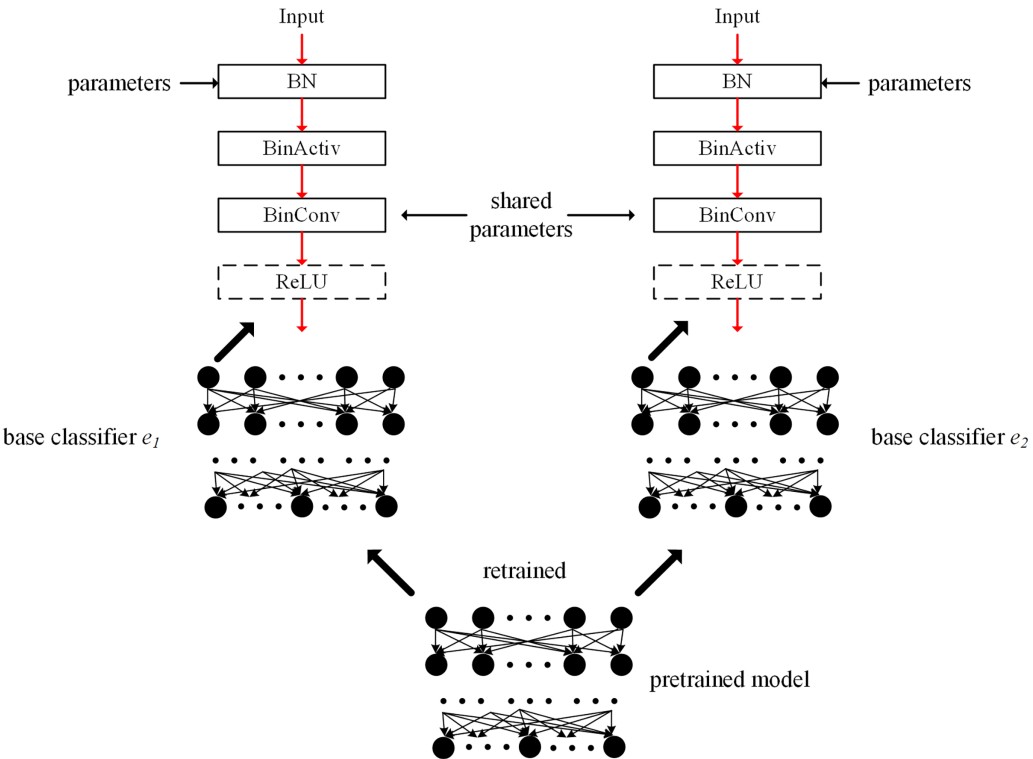

**Figure 2  Conceptual figure of proposed ensemble-based system using BNNs.**

in a classifier. If other base classifiers produce smaller quantization errors for their activations, it is assured that the problem of the maximized quantization error can be mitigated. The dot products for each base classifier maintain their optimization shown in Eq. (8). Besides, each base classifier has different learnable weights of last fully connected layer, adjusting the accumulation of the final features.

Algorithm 1 formally describes the retraining process mentioned above. A pretrained model $BNN_{pretrained}$ is used to initialize multiple $M$ base classifiers $BNN_{base}(M)$. A function Freeze($BinConv$) prevents updating the weights of $BinConv$ layers in all base classifiers. During training $T$ steps, the ensemble-based system with $M$ base classifiers is trained. A function GetWeights($BNN_{base}(M)$) produces the weights of the retrained ensemble. A function RemoveOverlap($weights$) removes the overlapped weights of $BinConv$ layers between base classifiers. Finally, the trained $weights$ are returned.

For better classification results, we can choose which filters are shared or not. Figure 3 illustrates basic blocks of the binarized ResNet (*He et al., 2016*). In the binarized ResNet, the basic blocks are stacked, maintaining a pyramid structure. The non-zero stride can reduce the resolution of output features by their height and width. When the number of channels is doubled in the ResNet, *stride* = 2 is adopted in the convolution.

In Fig. 3A, a basic block contains the shortcut summing the input features to the output of the last BN layer. The heights and widths of input and output features are the same, respectively. The basic block of Fig. 3A contains two BinConv layers. In this case, it is

| Algorithm 1 | Training of ensemble-based system using BNNs. |
|---|---|

1:  **procedure** TRAINING(pretrained model $BNN_{pretrained}$, training dataset *dataset*, number of base classifiers $M$, number of training steps $T$)

2:    $BNN_{base}(M) \Leftarrow$ Initialize($BNN_{pretrained}$, $M$)

3:    **for** $BinConv \in BNN_{base}(M)$ **do**

4:        Freeze($BinConv$)

5:    **end for**

6:    **for** $k \Leftarrow 1$ to $T$ **do**

7:        $BNN_{base}(M) \Leftarrow$ Train($BNN_{base}(M)$)

8:    **end for**

9:    *weights* $\Leftarrow$ GetWeights($BNN_{base}(M)$)

10:    *weights* $\Leftarrow$ RemoveOverlap(*weights*)

11:    **return** weights

12:  **end procedure**

possible that only one of them can share its filter weights in an ensemble-based system. In Fig. 3B, when *stride* = 2, the height and width of output features are half of those of input features. An 1 × 1 exact Conv layer can be used in the shortcut when shrinking the feature dimension. In another ensemble-based system, filter weights for this 1 × 1 Conv layer may not be shared between base classifiers.

## HARDWARE ANALYSIS

### Storage resource requirements

The filter sharing in the proposed ensemble-based systems reduces the storage resource requirements. For example, the binarized ResNet-20 structure for the CIFAR dataset is shown in Fig. 4. The conv1 and fully-connected linear blocks adopt precise fp32 operations like *Rastegari et al. (2016)*. The layer1, layer2, and layer3 blocks contain six 3 × 3 BinConv layers, respectively, where a basic block of the dotted box contains two BinConv layers. Each basic block contains the shortcut, which is described as the round red arrow. The dotted round red arrows mean 1 × 1 exact convolutional layers used as the shortcut for shrinking the feature dimension with *stride* = 2. Finally, the average pooling layer (denoted as *Avg pooling*) averages the final convolutional outputs. The linear layer has full connections to all averaged outputs to produce the final classification result.

Table 1 lists the output size, layer description, and storage requirements. The weight size of each binarized convolutional layer can be calculated as $c_{in} \times w \times h \times c_{out}$ bits. On the other hand, the weight size of the first fp32 convolutional layer in conv1 block is calculated as $c_{in} \times w \times h \times c_{out} \times 32$ bits. In the linear layer, $c_{out}$ can be the same as the number of classes. The storage requirements for the linear layer increase with the number of classes. For example, the storage requirements of the linear layer for the CIFAR-100 dataset can be 204,800 bits, providing 100 image classes.

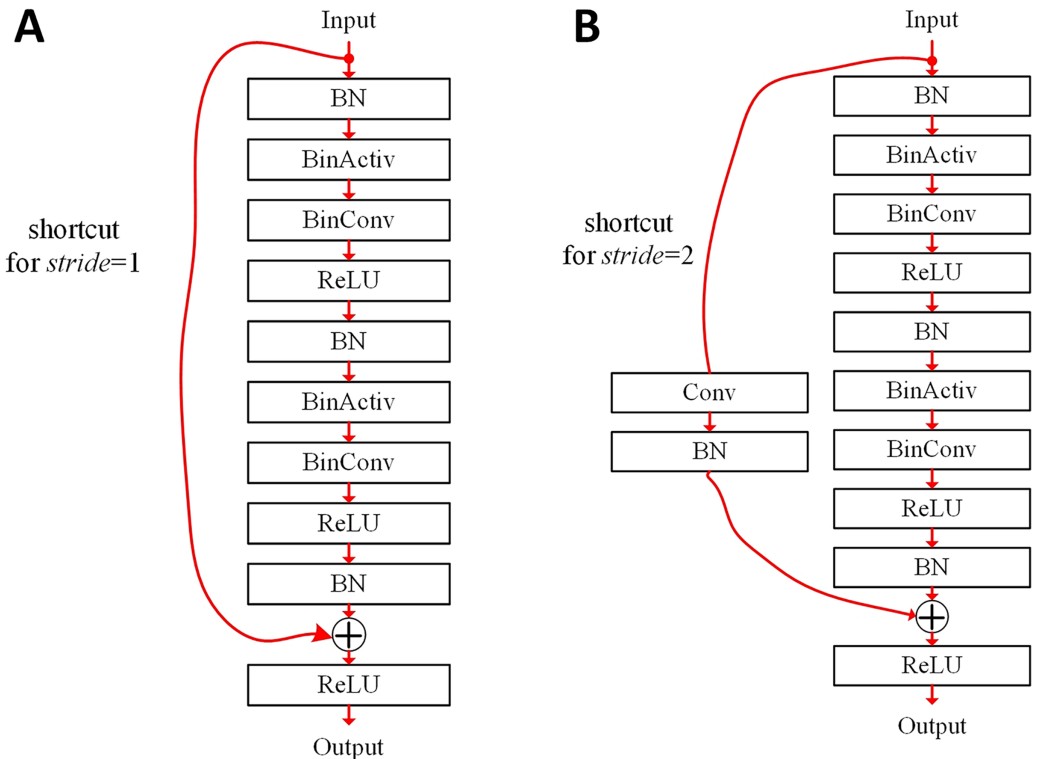

**Figure 3** Basic blocks of binarized ResNet (*He et al., 2016*): **(A)** *stride = 1*; **(B)** *stride = 2*.

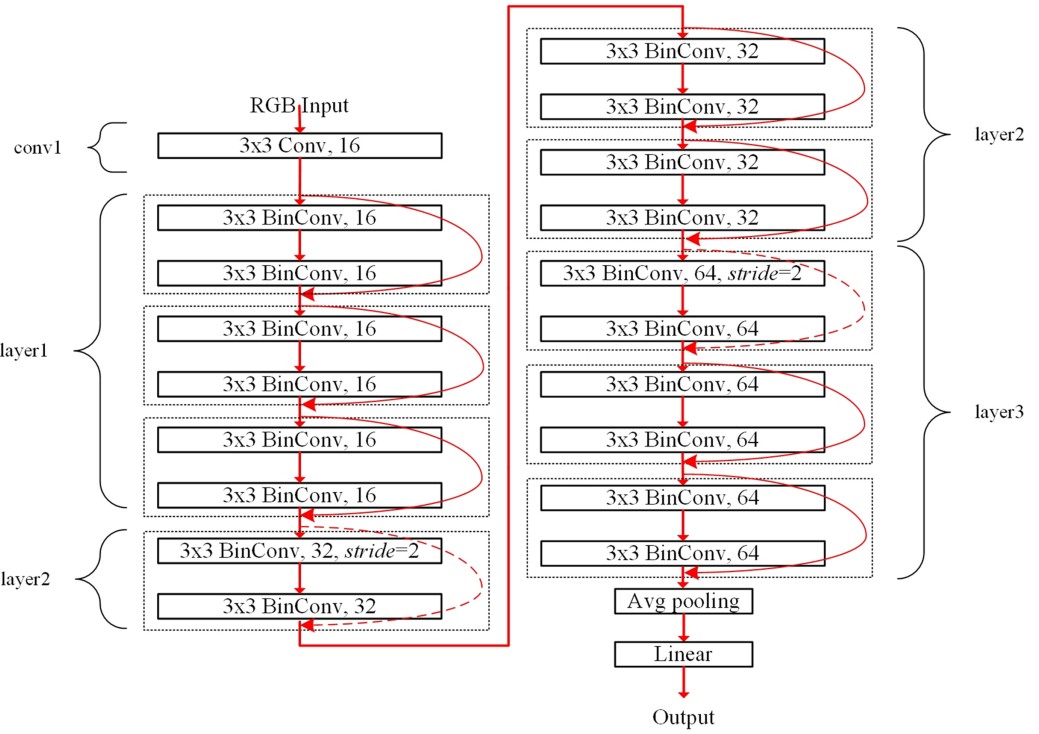

**Figure 4** Binarized ResNet-20 structure for the CIFAR dataset.

**Table 1 Details of binarized ResNet-20 model and storage resource requirements.**

| Block name | Output size[a] | Layer description[b] | Storage requirements (bits)[c] |
|---|---|---|---|
| conv1 | $16 \times 32 \times 32$ | $3 \times 3$, 3, *stride* = 1 | 13,824 |
| layer1 | $16 \times 32 \times 32$ | binarized $3 \times 3$, 16, *stride* = 1 | 13,824 |
| layer2 | $32 \times 16 \times 16$ | binarized $3 \times 3$, 16, *stride* = 2 | 67,072 |
| layer3 | $64 \times 8 \times 8$ | binarized $3 \times 3$, 32, *stride* = 2 | 268,288 |
| average pooling | $1 \times 1 \times 64$ | $8 \times 8$ average pooling | – |
| linear | 10 (CIFAR-10) | $1 \times 1$, 64, no stride | 20,480 |

Notes:
[a] When the number of output channels and the width and height of output features are denoted as $c_{out}$, $w_{out}$, and $h_{out}$, the output size is calculated as $c_{out} \times w_{out} \times h_{out}$.
[b] Terms denote the weight filter size $w \times h$, the number of input channels $c_{in}$, and the stride used in the first convolutional layer of the basic block. When *stride* = 2, $c_{out} = c_{in} \times 2$.
[c] Memory requirements for storing weights.

## Computational resources and power consumption

The ensemble-based system using these homogeneous base classifiers can increase computations proportional to the number of base classifiers in the inference stage. Although filter weights are shared, each base classifier follows the same binarized CNN structure, performing the same number of multiply-accumulate operations when using fusion, voting, bagging, and boosting schemes. For example, in *Sagartesla (2020)* and *Kim (2021)*, the binarized ResNet-18 for the CIFAR-10 dataset requires $58.6 \times 10^7$ floating-point operations (FLOPs). When $M$ base classifiers perform $M \times 58.6 \times 10^7$ FLOPs. Power consumption is also proportional to the number of base classifiers. For example, in *Guo et al. (2021)*, the estimated power consumption of the XNOR-Net model for ImageNet dataset (*Deng et al., 2009*) is 1.92 mJ. In this case, if $M$ base classifiers are adopted, power consumption can be $M \times 1.92$ mJ.

# EXPERIMENTAL RESULTS AND ANALYSIS

## Binarized ResNets on CIFAR datasets

We evaluated the binarized ResNet-20 and ResNet-18 models on the CIFAR datasets. The reasons why we chose these models and datasets are explained as follows. The skip connection or shortcut originated from ResNet (*He et al., 2016*) has been used in many prominent CNN models. Because the shortcut can pass the exact features to the next layer without information loss, BNNs using the shortcut could outperform the binarized plain CNN model. In *Rastegari et al. (2016)*, compared with the binarized plain CNN model, the performance of the binarized ResNet model was better. The outstanding benefits of the binarized ResNet model were considered in our experiments. Whereas the binarized ResNet-20 model in Fig. 4 and Table 1 was a simple lightweight BNN, the binarized ResNet-18 model required additional computational and storage resources. Table 2 lists details of binarized ResNet-18 model and storage resource requirements. Compared with the binarized ResNet-20 in Table 2, the storage requirements of the binarized ResNet-18 were about 47 times more than those of the binarized ResNet-20. Besides, the binarized ResNet-18 model required 13.5 times more computational resources.

**Table 2 Details of binarized ResNet-18 model and storage resource requirements.**

| Block name[a] | Output size[a] | Layer description | Storage requirements (bits) |
|---|---|---|---|
| conv1 | $64 \times 32 \times 32$ | $3 \times 3$, 3, *stride* = 1 | 5.53E+4 |
| layer1 | $64 \times 32 \times 32$ | binarized $3 \times 3$, 64, *stride* = 1 | 1.47E+5 |
| layer2 | $128 \times 16 \times 16$ | binarized $3 \times 3$, 64, *stride* = 2 | 7.78E+5 |
| layer3 | $256 \times 8 \times 8$ | binarized $3 \times 3$, 128, *stride* = 2 | 3.11E+6 |
| layer4 | $512 \times 4 \times 4$ | binarized $3 \times 3$, 256, *stride* = 2 | 1.25E+7 |
| average pooling | $1 \times 1 \times 64$ | $8 \times 8$ average pooling | – |
| linear | 10 (CIFAR-10) | $1 \times 1$, 512, no stride | 1.64E+5 |

Note:
[a] Layer1, layer2, layer3, and layer4 blocks contain two basic blocks, respectively.

In the CIFAR dataset (*Krizhevsky, Nair & Hinton, 2014*), 50,000 $32 \times 32$ training color images and 10,000 $32 \times 32$ test color images are prepared in the training and inference, respectively. After finishing retraining, trained models were evaluated with the test images. Whereas the CIFAR-10 dataset contains 10 different classes, the CIFAR-100 dataset has 100 classes. Because more sophisticated classification is needed, the classification accuracy on the CIFAR-100 dataset could be lower than that on the CIFAR-10 when adopting the same BNN structure.

## Evaluations of ensembles with binarized ResNets

An ensemble using the binarized ResNet-20 model was experimented on the CIFAR-100 dataset to know the performance and scalability of the proposed ensemble-based system. Firstly, the binarized ResNet-20 model was evaluated on the CIFAR-100 dataset. The initial weights of this model were trained, which is denoted as $BNN_{pretrained}$ in Algorithm 1. The training setup for producing the initial weights was as follows: In the data augmentation of input images, a $32 \times 32$ input image was randomly cropped from a $40 \times 40$ padded image and randomly flipped in the horizontal direction. However, this data augmentation above was not applied in the inference. We ran the training for 400 epochs with a batch size of 256. This training adopted the default Adam optimizer (*Kingma & Ba, 2014*) with betas = (0.9, 0.999). It was known that the error of binarized operations could be helpful in regularizing BNN models so that the regularization decaying updated weights was not adopted by zero weight decay. The learning rate was changed depending on the *poly* policy so that the learning rate *lr* decreased by $base_{lr} \times (1 - \frac{iteration}{epochs})$. The term $base_{lr}$ means the starting learning rate, which was initialized as 0.001. For the CIFAR-100 dataset, the dropout (*Srivastava et al., 2014*) layer with *dropout* = 0.5 was inserted just before the linear fully-connected layer. Parametric Rectified Linear Unit (PReLU) (*He et al., 2015*) layers were used in the activation like *Gu et al. (2019)*, *Phan et al. (2020)* and *Martinez et al. (2020)*.

Several ensemble schemes were experimented with using the Ensemble Pytorch library in *Xu (2020)*. In these evaluations, all filter weights of the binarized convolutional layers in each base classifier were initialized using the trained initial weights and then frozen during the training process. Regardless of the number of base classifiers *M*, the filter

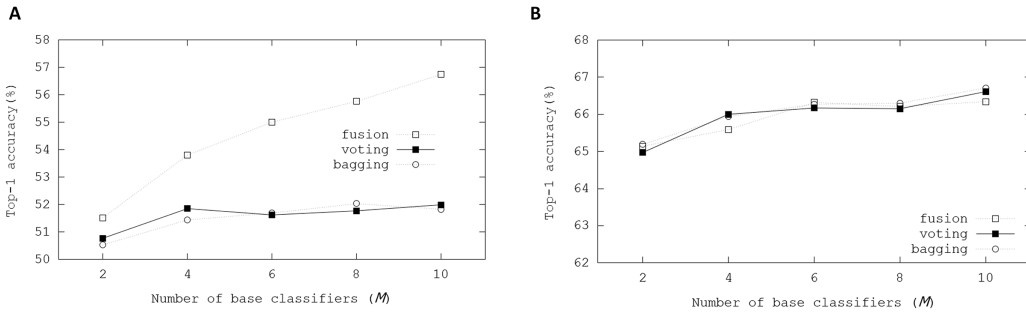

**Figure 5 Top-1 inference accuracies of ensemble schemes using binarized ResNet models on CIFAR-100 dataset: (A) binarized ResNet-20; (B) binarized ResNet-18.**

weights were shared between classifiers. On the other hand, the learnable parameters in the BN layer and weights of the linear layers were retrained in the ensemble.

The retraining of this ensemble was performed for 200 epochs with a batch size of 256 and adopted the Adam optimizer with betas = (0.9, 0.999). The starting learning rate was initialized as 0.01. Experimental results of the boosting and snapshot ensembles showed significantly degraded Top-1 accuracies. For example, when $M = 2$, the boosting scheme achieved 35.68% Top-1 accuracy. Therefore, we did not consider the boosting and snapshot ensemble schemes in the following. Instead, the fusion, voting, bagging schemes were evaluated. Our codes have been available at https://github.com/analog75/ensemble_bnn.

Figure 5A illustrates Top-1 inference accuracies of ensemble-based systems using trained binarized ResNet-20 on the CIFAR-100 dataset. While accuracies of the fusion ensemble scheme were proportional to $M$, those of the voting and bagging schemes cannot increase with $M \geq 4$. Whereas real-valued ResNet-20 can produce 64.26% Top-1 accuracy, the binarized ResNet-20 achieved 49.16% Top-1 accuracy. On the other hand, the fusion scheme got 56.74% Top-1 accuracy when $M = 10$. These evaluation results showed that the ensemble scheme can enhance the inference accuracy.

In other experiments, the binarized ResNet-18 on the CIFAR-100 dataset was evaluated to know the effectiveness of ensemble schemes on more complex BNN model. Whereas real-valued ResNet-18 achieved 75.61% Top-1 accuracy, Top-1 accuracy from the trained binarized ResNet-18 was 68.59%. Base classifiers shared the weights of the convolutional layers in the ensemble. The learnable parameters in the BN layer and weights of the linear layers were retrained in the ensemble. The retraining of this ensemble was performed for 200 epochs with a batch size of 256 except for the fusion scheme. Besides, the Adam optimizer was adopted with betas = (0.9, 0.999). When retraining the ensembles, two Tesla V100 graphic processing units (GPU) of Nvidia (*Cui et al., 2018*) were used in our GPU-based workstation. When $M \geq 8$ in the fusion ensemble method, GPU memory requirements exceeded the maximum resources of our workstation with a batch size of 256. Therefore, for the fusion ensemble scheme, the binarized ResNet-18 was evaluated with a batch size of 128.

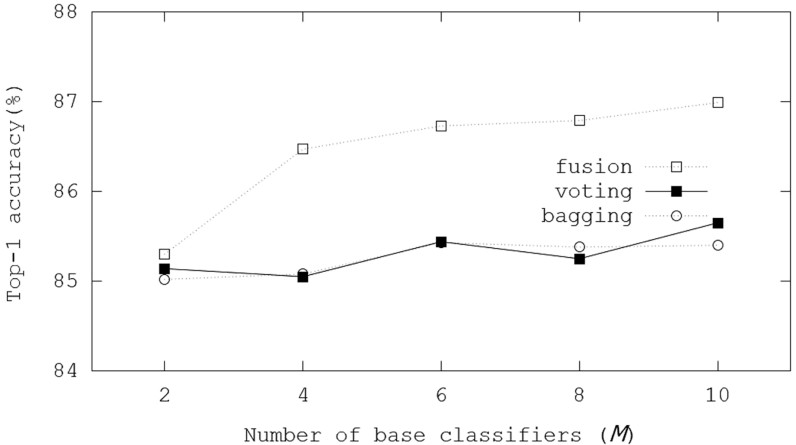

**Figure 6 Top-1 inference accuracies of ensemble schemes using binarized ResNet-20 models on the CIFAR-10 dataset.**

These ensemble schemes showed enhanced performances in the experimental results compared with the results with the binarized ResNet-20. The bagging ensemble scheme explicitly showed the scalability with $M$, providing small performance enhancements. When $M = 10$, the bagging ensemble scheme achieved 70.21% Top-1 inference accuracy. Compared with the evaluations using ResNet-20, the enhancements using the ensemble schemes were not significant in those when using ResNet-18. The initial weights of ResNet-18 produced higher accuracy than those of ResNet-20, meaning that the margin that can be improved using ensemble schemes was small with the binarized ResNet-18.

The CIFAR-10 dataset was adopted in ResNet-20 to know the effect of the number of classes. Although images of the CIFAR dataset were the same, the numbers of classes for the CIFAR-10 were only 10. In the data augmentation, from each 40 × 40 padded image, 32 × 32 input image was cropped and randomly flipped in the horizontal direction. The training ran for 400 epochs with a batch size of 256. Also, we used the Adam optimizer. The regularization decaying updated weights was not adopted. The initial learning rate was 0.001, changing with the *poly* policy. For the CIFAR-10 dataset, this training did not insert the dropout layers. The PReLU (*He et al., 2015*) was used in the activation of each basic block.

In the initialization, a trained binarized ResNet-20 model was adopted, where the binarized ResNet-20 model achieved 84.06% Top-1 accuracy for the CIFAR-10 dataset. Figure 6 illustrates Top-1 inference accuracies of ensemble-based systems using trained binarized ResNet-20 on the CIFAR-10 dataset. In the evaluations, the fusion scheme got 85.30% Top-1 inference accuracy with $M = 2$, and 86.99% Top-1 inference accuracy was achieved with $M = 10$. Compared with the case for the CIFAR-100 dataset, the enhancements with increasing $M$ were not significant so that we concluded that accuracy margins to be improved were small. However, the evaluation results showed the scalability with $M$. Besides, about 3% Top-1 inference accuracy was enhanced with $M = 10$, compared with the inference only using the initial BNN model.

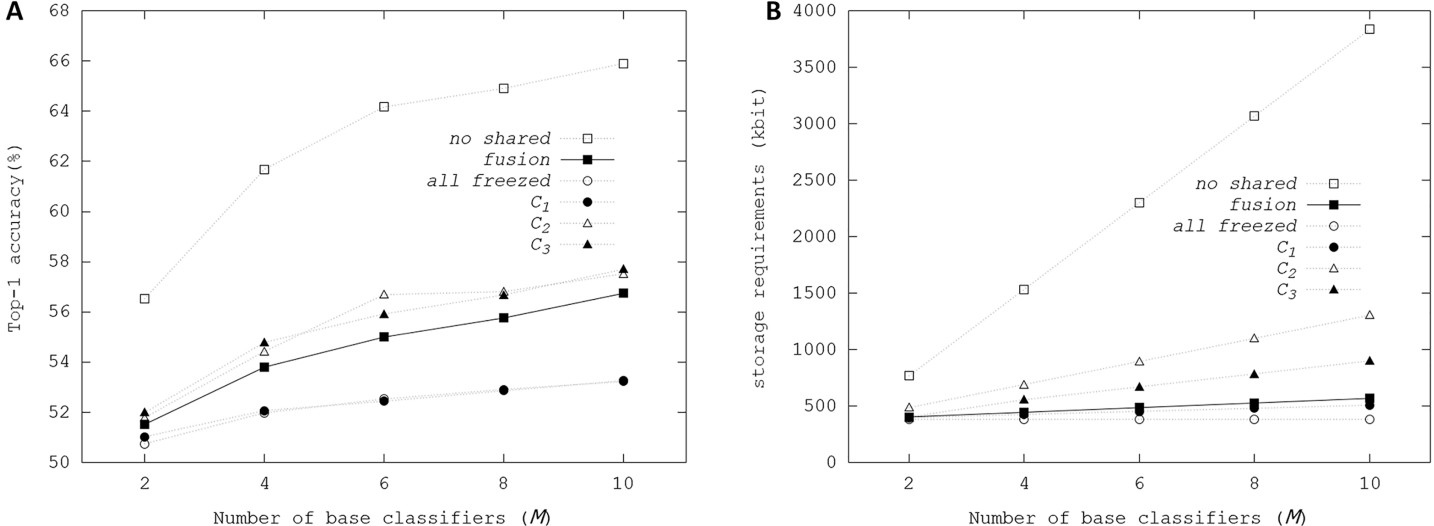

**Figure 7 Top-1 inference accuracies and storage requirements of different configurations of ensembles using binarized ResNet-20 on CIFAR-100 dataset: (A) Top-1 inference accuracy; (B) storage resource requirements.**

## Comparison on different configurations of weight sharing

We compared different configurations of ensembles depending on shared weights, proving that the ensemble using shared weights enhanced accuracies.

The other configurations are described as:

- *no shared*: $M$ base classifiers did not share weights in any layers.
- *fusion*: base classifiers shared the weights of all convolutional layers and did not share the weights of the linear layer.
- *all frozen*: base classifiers shared the weights of all convolutional and linear layers.
- $C_1$: except for the first convolutional layer, base classifiers shared the weights of all convolutional and linear layers.
- $C_2$: except for $1 \times 1$ convolutional layers used as shortcuts and linear layer, base classifiers shared the weights of all convolutional layers.
- $C_3$: except for the last convolutional and linear layers, base classifiers shared the weights of all convolutional layers.

Figure 7 illustrates Top-1 final accuracies and storage resource requirements depending on the configurations of shared weights using the fusion ensemble scheme. In *no shared* with $M = 10$, Top-1 accuracy reached up to 65.9%, showing that large storage resources can increase performance in the ensemble. When the weights of all convolutional and linear layers were shared in the *all frozen* configuration, Top-1 final accuracies showed the scalability with $M$. Notably, when $M = 10$, Top-1 final accuracy was 53.23%, increasing by 4%, compared with 49.16% Top-1 accuracy from the initial weights. Whereas all weights of convolutional and linear layers were shared in *all frozen* configuration, the affine parameters of BN layers were trainable, producing the accuracy enhancements

proportional to $M$. The memory requirements for storing multiple affine parameters in the ensemble were tiny so that the costs for increasing storage resource requirements were negligible. When the weights of the first convolutional layer were not shared in $C_2$ configuration, there were no performance enhancements over the *fusion* and $C_1$ configurations. We concluded that when base classifiers did not share the filters of the first convolutional layer, the performance enhancements could be negligible in the ensemble-based system. On the other hand, $C_2$ and $C_3$ configurations had slight enhancements over the *fusion* configuration. However, Fig. 7B shows that storage resource requirements of the $C_2$ and $C_3$ increased sharply with $M$.

## Ensembles with Bi-Real-Net and ReActNet on CIFAR dataset

Among several recent BNN structures, Bi-Real-Net (*Liu et al., 2018*) was one of the outstanding works, proposing the shortcut for each convolutional layer. We modified the original code of Bi-Real-Net for the evaluation on the CIFAR-100 dataset. Data augmentation process was the same as the method mentioned for the binarized ResNet-18. Like the binarized ResNet-18, Bi-Real-Net-18 contained 16 binarized convolutional layers. The first convolutional and linear layers got real-valued features as inputs for achieving high classification accuracy. The downsampling was performed with $1 \times 1$ real-valued convolutional layer per four binarized convolutional layers. The training process for producing the initial weights was as follows: The training was performed for 200 epochs with a batch size of 256. The Adam optimizer was applied with betas = (0.9, 0.999). Regularization methods using the dropout layer and non-zero weight decay were not adopted. The initial learning rate is 0.001, changing with *poly* policy. With the trained weights, Bi-Real-Net-18 can achieve 63.97% Top-1 final accuracy on setups mentioned above.

Fusion, soft voting, and bagging schemes were applied to evaluate ensemble-based systems using Bi-Real-Net-18. The affine parameters of the BN layers and weights of linear layer for each base classifier were retrained. During the retraining with base classifiers, the initial learning rate was 0.001, and the learning rate was changed according to *poly* policy. The retraining was performed for 200 epochs with a batch size of 256. Also, the Adam optimizer was used. Like the cases using binarized ResNet-18, the batch size was setup as 128 for fusion schemes.

Figure 8 illustrates Top-1 inference accuracies of ensemble-based systems using trained Bi-Real-Net-18 on the CIFAR-100 dataset. In the evaluations, the voting and bagging schemes performed better than the fusion scheme. The number of channels in each layer of Bi-Real-Net-18 is equal to that of ResNet-18. Top-1 inference accuracies were enhanced by 2.74–1.17%. Notably, the bagging scheme achieved 66.71% Top-1 inference accuracy with $M = 10$. The performance improvement became small as $M$ increased like the cases of binarized ResNet-18, but the evaluation results showed increasing accuracies with $M$.

On the other hand, ReActNet (*Liu et al., 2020*) was the-state-of-the-art BNN model, where ReAct-Sign (RSign) and ReAct-PReLU (RPReLU) were proposed to reshape and shift activation distribution. We evaluated our ensemble methods by sharing filters of binarized convolutional layers on the ReActNet-10 model. Like the binarized ResNet-10,

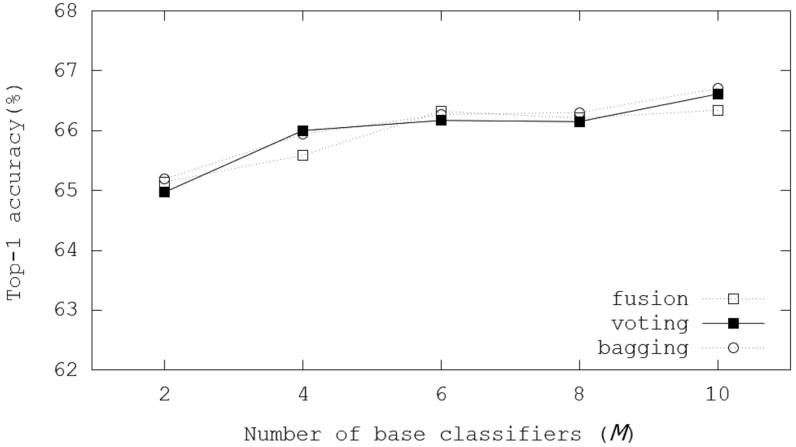

**Figure 8 Top-1 inference accuracies of ensemble schemes using Bi-Real-Net-18 on the CIFAR-100 dataset.**

ReActNet-10 contained eight binarized convolutional layers, where each convolutional layer has its own shortcut and is connected with Rsign and RPReLU layers. The downsampling with $1 \times 1$ binarized convolutional layer was performed per two binarized convolutional layers. The training process for producing the initial weights was as follows: Firstly, real-valued trained model is obtained. Then, the training was performed for 400 epochs with a batch size of 256 and zero weight decay. The Adam optimizer was applied betas = (0.9, 0.999). The initial learning rate is 0.0005, changing with *poly* policy. In our evaluation, the trained ReActNet-10 model achieved 66.69% Top-1 final accuracy on the CIFAR-100 dataset.

With the pretrained ReActNet-10 model, fusion, soft voting, and bagging schemes were applied to evaluate ensemble-based systems. Like ResNet and Bi-Real-Net, the affine parameters of the BN layers and weights of linear layer for each base classifier were retrained. Besides, our experiments retrained parameters for controlling the thresholds of RSign layers. For RPReLU layers, several parameters for moving value distributions and controlling the slopes of negative parts were retrained. On the other hand, the filter weights of binarized convolutional layers were shared in ensemble-based systems. Weights of downsampling layers were not shared. During the retraining with base classifiers, the initial learning rate was 0.001, and the learning rate decreased according to *poly* policy. The retraining was performed for 200 epochs with a batch size of 256. The Adam optimizer was used. The retraining of this ensemble was performed for 200 epochs with a batch size of 256.

Figure 9 illustrates Top-1 inference accuracies of ensemble-based systems using trained ReActNet-10 on the CIFAR-100 dataset. In the evaluations, Top-1 inference accuracies were enhanced by 3.6–1.7%. Notably, the bagging scheme got 70.29% Top-1 inference accuracy with $M = 10$, which enhanced Top-1 final accuracy by 3.6%. Like results of the binarized ResNet-18 in Fig. 5, the bagging scheme was the best in our evaluations, where 70.29% Top-1 inference accuracy was achieved with $M = 10$. The evaluation results showed the scalability with $M$. The performance improvement slowed down as $M$

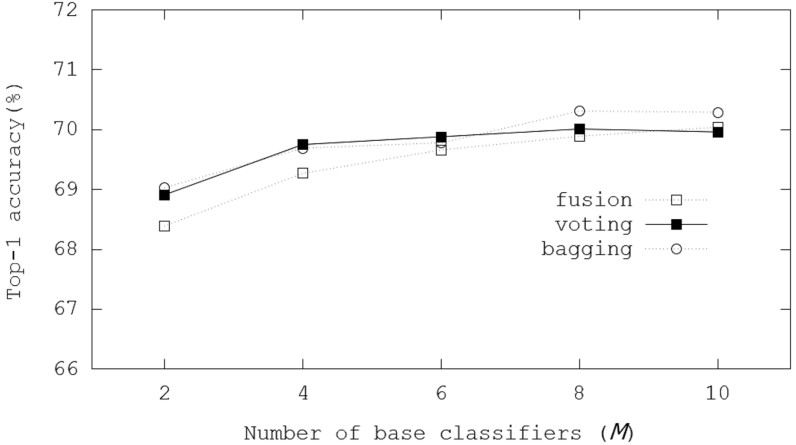

**Figure 9 Top-1 inference accuracies of ensemble schemes using ReActNet-10 on the CIFAR-100 dataset.**

increased, but the ensemble schemes always provided better performance over the case only using one classifier.

## CONCLUSION

The proposed ensemble-based system using shared filter weights can reduce storage resource requirements and shows the scalability with the number of base classifiers. In different scenarios of filter sharings, the trained ensembles can enhance the classification accuracies and show trade-off relationships between storage requirements and classification performance. We evaluated the proposed method on the-state-of-the-art BNN models and described the detailed training process, proving that our storage-efficient ensemble can enhance classification accuracies. Most of all, it was concluded that the proposed method can provide a scalable solution and extend applications of ensemble-based systems using BNN models.

### Funding

The EDA tool was supported by the IC Design Education Center (IDEC), Korea. This research was the result of a study on the "HPC Support" Project, supported by the 'Ministry of Science and ICT' and NIPA. This work was supported by the National Research Foundation of Korea (NRF) grant funded by the Korea government (MSIT) (No. 2021R1F1A1048054). The funders had no role in study design, data collection and analysis, decision to publish, or preparation of the manuscript.

### Grant Disclosures

The following grant information was disclosed by the authors:
IC Design Education Center (IDEC), Korea.
'Ministry of Science and ICT' and NIPA.
Korea Government (MSIT): 2021R1F1A1048054.
## Competing Interests

The authors declare that they have no competing interests.

## Author Contributions

- HyunJin Kim conceived and designed the experiments, performed the experiments, analyzed the data, performed the computation work, prepared figures and/or tables, authored or reviewed drafts of the paper, and approved the final draft.
- Mohammed Alnemari performed the experiments, analyzed the data, authored or reviewed drafts of the paper, important comments and idea discussion, and approved the final draft.
- Nader Bagherzadeh analyzed the data, authored or reviewed drafts of the paper, important comments and idea discussion, and approved the final draft.

## Data Availability

The raw data is available at GitHub: https://github.com/analog75/ensemble_bnn.

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
