# Peer review of "A storage-efficient ensemble classification using filter sharing on binarized convolutional neural networks"

_PeerJ Computer Science, doi:10.7717/peerj-cs.924_

## Round 0.1 · original submission · Major Revisions

Based on the reviewers' comments, the manuscript needs more efforts and work in order to be in a good shape for being accepted.

Also, two of the reviewers have requested that you cite specific references. You may add them if you believe they are especially relevant. However, I do not expect you to include these citations, and if you do not include them, this will not influence my decision.

Reviewer 1 ·

Basic reporting

The authors submitted a very interesting work, my suggestions to improve it are:
1.I suggest to add it to github “The source code for reproducing the experiments will be available upon publication of the manuscript”
2.The contribution is no introduced clearly on the theoretical analysis; also if very good results are reported. If feasible add some more details on the theoretical analysis of your approach.
3.You should better review the literature on ensemble of classifiers, e.g.
https://arxiv.org/abs/1802.03518
https://doi.org/10.1016/j.eswa.2020.114048
https://arxiv.org/pdf/2104.02395.pdf
4.Some typos, E.g. row 153 “BNN model can be used in a low poer” -> “...power”

Experimental design

5.You run many experiments and you have reported many results, this is appreciated, anyway please better stress the novelty of your method respect the literature on pruning and quantization approaches:
https://www.sciencedirect.com/science/article/pii/S0031320321000868
https://arxiv.org/pdf/2103.13630.pdf

Validity of the findings

no comment

Reviewer 2 ·

Basic reporting

1.The last word on line 153 is wrong. Suggest to examine each word and sentence carefully.
2. Suggest to polish the language of the writing.

Experimental design

No comment

Validity of the findings

Figure 5(b) does not contained data when in the fusion ensemble due to the limitation of GPU resources. I suggest you to supplement the result when , in order to ensure the integrity of the experiment. You can use smaller batch size or use CPU for retraining.

Additional comments

In this manuscript, the authors proposed a storage-efficient ensemble classification to overcome the low inference accuracy of binary neural networks (BNNs). The work indicates that proposed method reduces the storage burden of multiple classifiers in the lightweight system. This is a good idea, which can be used to improve the accuracy and reduce the storage burden of BNN. In addition, this manuscript also provides a solution for the application of neural network in lightweight system. There are some suggestions as follows:
1. Fusion, voting, and bagging schemes were applied to evaluate ensemble-based systems. However, there is no comparison between these methods in the paper.
2. Your introduction at lines 296-310 needs more detail. I suggest that you can add figure to describe the experiment result.

Reviewer 3 ·

Basic reporting

This is the review report of the paper entitled "A storage-efficient ensemble classification using filter sharing on binarized convolutional neural networks".

The paper presents a very important topic and is well presented. However, I have some comments to improve the paper.

1- In the abstract, add the value of classification accuracy to support the theory.

2- The authors show the results of their proposed method with the state-of-the-art model (ResNet), I would suggest showing the results of the ResNet on the same dataset without the use of the proposed method.

3- training parameters are required to mention.

4- Paper code with a nice demo is important to upload on any public platform.

5- I would suggest citing the following reference when referring to CNN so new reference from 2021 can be used
https://link.springer.com/article/10.1186/s40537-021-00444-8

6-Comparison with state-of-the-art is necessary to add on the same used dataset.

7-explain more on the research gap of previous methods.

8-The contributions of the article have to be clear for the readers, I would suggest making them as bullet points at the end of the introduction.

Experimental design

See the first box " Basic reporting"

Validity of the findings

See the first box " Basic reporting"

Additional comments

See the first box " Basic reporting"

---

## Round 0.2 · accepted · Accept

The authors have addressed all the comments and issues that the reviewers pointed out, so the paper is now in a proper shape to be accepted for publication.

Reviewer 1 ·

Basic reporting

Revision well done

Experimental design

Revision well done

Validity of the findings

Revision well done

Reviewer 2 ·

Basic reporting

No comment

Experimental design

No comment

Validity of the findings

No comment

Additional comments

No comment

Reviewer 3 ·

Basic reporting

The authors addressed the comments in a very good way.

Experimental design

The authors addressed the comments in a very good way.

Validity of the findings

The authors addressed the comments in a very good way.